# Instruction Tuning With Loss Over Instructions

**Zhengyan Shi**[1] **Adam X. Yang**[2] **Bin Wu**[1]
**Laurence Aitchison**[2] **Emine Yilmaz**[1] **Aldo Lipani**[1]
[1]University College London [2]University of Bristol
{zhengxiang.shi.19,bin.wu.23,emine.yilmaz,aldo.lipani}@ucl.ac.uk
{adam.yang,laurence.aitchison}@bristol.ac.uk

## Abstract

Instruction tuning plays a crucial role in shaping the outputs of language models (LMs) to desired styles. In this work, we propose a simple yet effective method, INSTRUCTION MODELLING (IM), which trains LMs by applying a loss function to the instruction and prompt part rather than solely to the output part. Through experiments across 21 diverse benchmarks, we show that, in many scenarios, IM can effectively improve the LM performance on both NLP tasks (*e.g.,* MMLU, TruthfulQA, and HumanEval) and open-ended generation benchmarks (*e.g.,* MT-Bench and AlpacaEval). Remarkably, in the most advantageous case, IM boosts model performance on AlpacaEval 1.0 by over 100%. We identify two key factors influencing the effectiveness of IM: (1) The ratio between instruction length and output length in the training data; and (2) The number of training examples. We observe that IM is especially beneficial when trained on datasets with lengthy instructions paired with brief outputs, or under the Superficial Alignment Hypothesis (SAH) where a small amount of training examples are used for instruction tuning. Further analysis substantiates our hypothesis that our improvement can be attributed to reduced overfitting to instruction tuning datasets. It is worth noting that we are not proposing IM as a replacement for the current instruction tuning process. Instead, our work aims to provide practical guidance for instruction tuning LMs, especially in low-resource scenarios. Our code is available at https://github.com/ZhengxiangShi/InstructionModelling.

## 1 Introduction

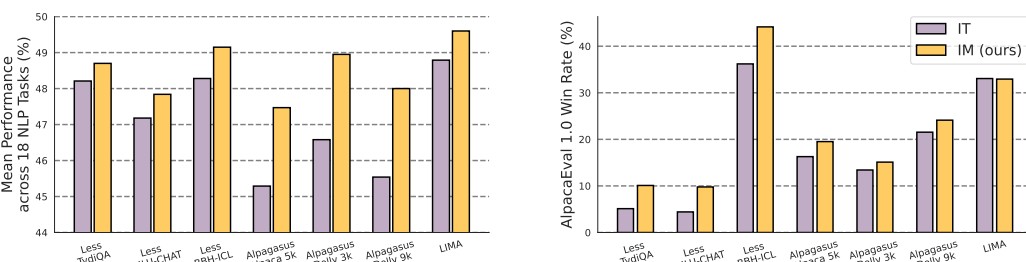

Figure 1: Performance differences between INSTRUCTION TUNING (IT) and our proposed method INSTRUCTION MODELLING (IM) trained on 7 instruction tuning datasets. These datasets contain prompts and responses but do not contain preference pairs. Specifically, we use the `Less` datasets [68] and `Alpagasus` datasets [11], which are subsets of `Flan V2` [14], `Dolly` [18], and `Stanford Alpaca` [61] to ensure good performance. We also report the results on the `LIMA` dataset. (**Left**) The mean performance across 18 traditional NLP tasks (see §4.1 for details). (**Right**) The win rate on the AlpacaEval 1.0 benchmark [37]. Please refer to §4.2 for details.

38th Conference on Neural Information Processing Systems (NeurIPS 2024).

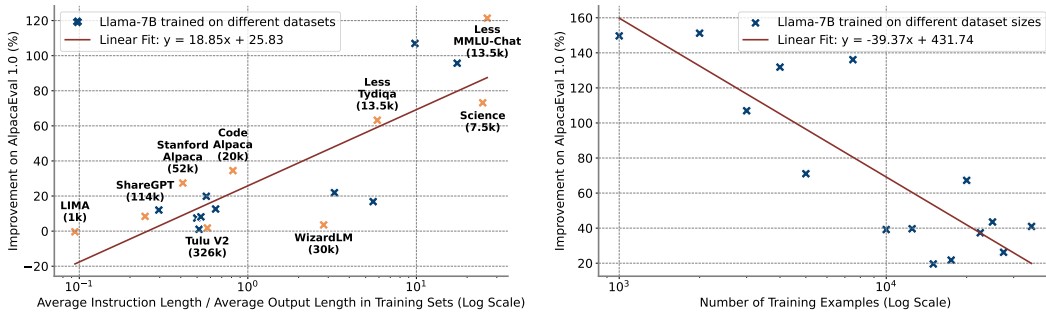

Figure 2: (**Left**) Performance improvement, achieved by our approach INSTRUCTION MODELLING (IM) compared to INSTRUCTION TUNING (IT) on the AlpacaEval 1.0, against the ratio between average instruction length and average output length in instruction tuning datasets (training size noted in parentheses). We highlight several representative instruction tuning datasets in yellow. Our analysis suggests that IM is especially beneficial for datasets characterized by lengthy instructions or prompts paired with comparably brief outputs, such as `Code Alpaca` [10] and `Less MMLU Chat` [68]. (**Right**) Performance improvement achieved by our approach IM over IT on the AlpacaEval 1.0 against the number of training examples in instruction tuning datasets. Here we maintain a fixed ratio between instruction and output length of 10. This analysis suggests that IM is particularly effective under the low-resource setting or Superficial Alignment Hypothesis. Please refer to §4.2 for details.

Language models (LMs) are trained to predict the next token on massive corpora, enabling them to learn general-purpose representations transferable to various language understanding or generation tasks [7, 47, 48, 56, 62]. However, it does not align LMs to act in accordance with the user's intentions [34]. To enable this transfer, various methods for aligning language models [3, 14, 44, 73, 58, 79] have thus been proposed, one of which is instruction tuning (IT) [36, 65, 69]. Recent study [78] proposes Superficial Alignment Hypothesis (SAH): A model's knowledge and capabilities are learnt almost entirely during pretraining, only minimal instruction tuning data is required to enable high-quality outputs in the desired output style. Existing works [1, 43, 44, 51, 65, 69, 36] mainly perform instruction tuning by focusing the loss computation solely on the output segments.

In this work, we demonstrate that in many scenarios, incorporating the loss computation for instructions or prompts, which we refer to as INSTRUCTION MODELLING (IM) (see §3), could substantially improve the performance of instruction tuning on both various NLP tasks (*e.g.,* MMLU, TruthfulQA, and HumanEval) and open-ended generation benchmarks (*e.g.,* MT-Bench and AlpacaEval), as shown in Figure 1. Remarkably, in the most favourable case, our proposed method IM boosts performance on AlpacaEval 1.0 by over 100%. As illustrated in Figure 2, our study further identifies two key factors influencing the effectiveness of IM: (1) **The ratio between instruction length and output length** (see Figure 2 Left). Our analysis shows that our approach IM is especially beneficial for datasets characterised by lengthy instructions or prompts paired with comparably brief outputs, such as `Code Alpaca` [13] and `Less MMLU Chat` [68]; (2) **The number of training examples** (see Figure 2 Right). We demonstrate that our approach IM performs better under the SAH, where a small amount of training examples are available (see §4.2).

Recent works [27, 31, 44, 71, 73] suggest that LMs can quickly memorise training examples even after seeing them just once. We hypothesise that the improvement stems from reducing instruction tuning's tendency to overfit, particularly under limited training resource conditions: Instruction tuning on brief outputs or a small amount of data can potentially lead to rapid overfitting. To substantiate our hypothesis, our analysis shows that (1) IM exhibits higher training losses but lower test losses on new instruction tuning data; (2) The outputs generated by IM have a lower similarity to the training examples compared to those from IT, as indicated by BLEU scores; and (3) IM leads to less performance degrade on NLP tasks across training epochs (see §4.3). Additionally, our study reveals that this overfitting cannot be effectively addressed by applying Kullback-Leibler (KL) divergence for regularisation [3, 44], as it compromises the model's ability to follow instructions. Our further analysis reveals that the advantages of IM persist across different LMs and model sizes, and that IM could be effectively combined with the previous approach (*i.e.,* NEFTUNE [31]). Meanwhile, we investigate the relationship between output length and win rate for our approach (see §4.4).

In summary, the main contributions of this paper are:

- We propose INSTRUCTION MODELLING (IM), aiming to enhance both the instruction-following and general performance on NLP tasks of LMs. Through extensive experiments across 21 benchmarks, we demonstrate that, in many scenarios, IM substantially improves the performance of LMs trained on various instruction tuning datasets, particularly notable in the AlpacaEval 1.0 benchmark where it boosts scores by over 100%.

- Our study identifies key factors influencing the effectiveness of IM, including the ratio between instruction length and output length and the number of training examples. We are not proposing IM as a replacement for current instruction tuning processes. Rather, we provide empirical guidance for fine-tuning LMs, especially under low-resource scenarios.

- We provide underlying mechanisms that make IM effective, specifically how it mitigates overfitting, thereby enhancing the LMs' performance across various tasks.

## 2   Related Work

**Instruction Tuning.**   LMs can better align with user intents through fine-tuning on datasets consisting of instructions and human-written completions [3, 44]. Early studies mainly focus on NLP tasks, showing that fine-tuning with various NLP datasets trained with instruction output pairs improves cross-task generalisation [1, 33, 43, 44, 51, 54, 65, 66]. Recent works explore the creation of instruction tuning datasets by LMs themselves [63, 25, 69, 36] or through crowdsourcing approaches [13, 78]. Such instruction-tuning phrase [30, 53, 59, 26, 73] enables LLMs to generalise beyond instructions in the training set, largely enhancing their practical utility.

**Data Selection for Instruction Tuning.**   Research on instruction tuning for LMs presents diverging perspectives on the optimal data scale for supervised fine-tuning. A prevailing view recommends fine-tuning on expansive datasets to enhance LM performance across various NLP tasks, thereby improving zero-shot and few-shot learning capabilities [1, 33, 44, 65, 43, 42, 51, 57, 64, 67]. For example, `Flan V2` comprises over a million question-answer pairs from diverse NLP sources [14], and `Natural Instructions` features 61 distinct tasks and 193k task instances [43]. Conversely, another research trajectory prioritises data quality over quantity [20, 70, 40, 32]. The Superficial Alignment Hypothesis (SAH) [78] advocates for using smaller, high-quality datasets, arguing that LMs primarily acquire their capabilities during the pretraining phase and thus require only minimal data for effective instruction tuning. For instance, LIMA [78] employs a carefully curated set of 1k diverse prompts to generate stylistically consistent responses, aimed at creating a helpful AI assistant. AlpaGasus [11] and LESS [68] employ methods to select high-quality data based on LLM-generated judgements and gradient signals. However, both views agree on the importance of (1) the quality of pre-trained base LMs and (2) the diversity and quality of the IT data.

**Regularisation Through Language Modelling Objectives.**   Pretraining data and language modelling objectives have been used as a regularisation technique in fine-tuning LMs. In particular, [15, 39] fine-tunes LMs on labelled data, with unsupervised learning on unlabelled data for auxiliary tasks as regularisation. [44] mixes the alignment objective with the next token prediction objective using pretraining data to mitigate alignment tax in reinforcement learning from human feedback (RLHF). [22] adopts the masked language objective on the pretraining or downstream task corpus to preserve pre-trained features, and shows improvements in calibration and accuracy. [29] investigates the effect of incorporating instruction loss weighting on instruction tuning, suggesting that the instruction loss ratio is an important hyperparameter when fine-tuning short-completion data but is irrelevant when using long-completion data. In this work, we propose a broader guideline that does not introduce new hyperparameters but focuses on when and how to include loss over instruction effectively. We refer to our approach as INSTRUCTION MODELLING because it combines elements of both language modelling and instruction tuning.

## 3   Our Approach

In this section, we first revisit the background of INSTRUCTION TUNING (IT) and then introduce our proposed method, INSTRUCTION MODELLING (IM).

**Instruction Tuning.**  In instruction tuning, each input is a concatenation of an instruction $I$ and a completion $C$. Let $I$ be the instruction sequence $\{I_1, I_2, \ldots, I_m\}$ and $C$ be the completion (output) sequence $\{C_1, C_2, \ldots, C_n\}$, where $I$ may include special prompt template tokens (such as "`<|user|>`" and "`<|assistant|>`"). The total input sequence $x$ is $\{I_1, I_2, \ldots, I_m, C_1, C_2, \ldots, C_n\}$. The model predicts each token in $C$ given all the previous tokens in $I$ and $C$ up to that point:

$$P(C_1, C_2, \ldots, C_n | I_1, I_2, \ldots, I_m) = \prod_{j=1}^{n} P(C_j | I_1, I_2, \ldots, I_m, C_1, C_2, \ldots, C_{j-1}) \qquad (1)$$

The loss function, $\mathcal{L}$, for instruction tuning is the negative log-likelihood of the completions given the instructions, expressed as follows:

$$\mathcal{L} = -\log P(C_1, C_2, \ldots, C_n | I_1, I_2, \ldots, I_m) = -\sum_{j=1}^{n} \log P(C_j | I_1, I_2, \ldots, I_m, C_1, C_2, \ldots, C_{j-1}) \qquad (2)$$

This approach aims to optimise the predictions for the completion sequence $C$, using the instruction sequence $I$ as contextual information.

**Our Approach: Instruction Modelling.**  Our approach, instruction modelling, is an expansion of instruction tuning by incorporating loss calculation for both the instruction and the completion tokens, except it omits any special prompt template tokens. The model is trained to predict both the instruction and completion parts of $x$ but excludes tokens that are part of prompt templates (denoted as $T$). For simplicity, we consider that these template tokens are not part of $I$ or $C$. The model predicts the next token given all previous tokens (both instructions and completions up to that point):

$$P(x) = P(I_1, I_2, \ldots, I_m, C_1, C_2, \ldots, C_n) = \prod_{t=1}^{m+n} P(x_t | x_1, x_2, \ldots, x_{t-1}) \qquad (3)$$

The loss function, $\mathcal{L}$, for instruction modelling calculates the negative log-likelihood for both instruction and completion tokens, excluding any prompt template tokens. It is computed as follows:

$$\mathcal{L} = -\sum_{t=1}^{m+n} \log P(x_t | x_1, x_2, \ldots, x_{t-1}) \cdot \mathbf{1}(x_t \notin T), \qquad (4)$$

where $\mathbf{1}(x_t \notin T)$ is an indicator function that is 1 if $x_t$ is not a template token and 0 otherwise. This ensures that the loss is computed only over the meaningful tokens, not over the static template tokens. Our approach allows the model to improve its understanding of both the instructions and the completions while being sensitive to the context provided by both segments of the input sequence.

## 4  Experiments and Results

In this section, we evaluate the effectiveness of our proposed method INSTRUCTION MODELLING (IM) by comparing it with INSTRUCTION TUNING (IT) and other baselines on various datasets.

### 4.1  Experimental Setup

**Instruction Tuning Datasets.**  We assess our method, IM, across various instruction tuning datasets, detailed as follows: (1) `Stanford Alpaca` [61] (52 002 examples); (2) `Dolly` [18] (15 011 examples); (3) `Sharegpt` [13] (50 000 examples); (4) `Code Alpaca` [10] (20 022 examples); (5) `Science Literature` [30] (7 544 examples); (6) `WizardLM` [69] (30 000 examples); (7) `Tulu V2` [30] (326 181 examples). Additionally, we incorporate instruction tuning datasets under the low-resource setting or SAH: (8) `LIMA` [78] (1 030 examples); (9) `Less`[1] [68], where high-quality instruction tuning data are selected from `Flan V2` and `Dolly`. Here, we use the `Less MMLU Chat` (13 533 examples), `Less BBH ICL` (13 533 examples), and `Less Tydiqa` (13 533 examples); (10) `Alpagasus`[2] [11], which offers three subsets: `Alpagasus Dolly 3k` (2 996 examples), `Alpagasus Dolly 9k` (9 229 examples) selected from `Dolly`, and `Alpagasus Alpaca 5k` (5 305 examples) selected from `Stanford Alpaca`. See dataset details and statistical analysis in Appendix §A.

---

[1]https://github.com/princeton-nlp/LESS
[2]https://github.com/gpt4life/alpagasus

Table 1: Performance comparisons using 7 instruction tuning datasets with the LLAMA-2-7B on 6 categories of 18 traditional NLP tasks and 3 open-ended benchmarks with LLM as judgements. "IT" refers to INSTRUCTION TUNING. "IM" refers to INSTRUCTION MODELLING. Green and red arrows indicate performance changes against the baseline (IT).

| Method | NLP Benchmarks | | | | | | | LLM-based Evaluation | | |
| | Understanding & Knowledge | Multi-linguality | Commonsense Reasoning | Math&Code Reasoning | BBH | Safety & Helpfulness | Mean | MT-Bench | AlpacaEval 1.0 | AlpacaEval 2.0 |
|---|---|---|---|---|---|---|---|---|---|---|
| LLAMA-2-BASE | 63.91 | 61.99 | 75.86 | 13.32 | 38.80 | 42.03 | 49.32 | 1.16 | 0.01 | 0.01 |
| LLAMA-2-CHAT | 63.42 | 55.15 | 70.28 | 15.33 | 38.92 | 51.79 | 49.15 | 6.63 | 79.04 | 6.48 |
| *Alpagasus Alpaca 5k (5,305 training examples)* | | | | | | | | | | |
| IT | 64.98 | 57.24 | 66.06 | 8.93 | 26.80 | 47.74 | 45.29 | 3.62 | 16.29 | 2.46 |
| NEFTUNE | 65.18 | 56.88 | 66.45 | 10.24 | 29.53 | 45.46 | 45.62↑0.33 | 3.50↓0.12 | 21.37↑5.08 | 2.37↓0.09 |
| IM (ours) | 64.01 | 56.63 | 72.47 | 11.58 | 35.52 | 44.62 | 47.47↑2.18 | 3.48↓0.14 | 19.52↑3.23 | 3.29↑0.83 |
| *Alpagasus Dolly 3k (2,996 training examples)* | | | | | | | | | | |
| IT | 65.81 | 57.46 | 67.55 | 11.96 | 33.02 | 43.70 | 46.58 | 4.23 | 13.42 | 2.00 |
| NEFTUNE | 65.90 | 57.79 | 67.28 | 11.64 | 35.43 | 44.36 | 47.07↑0.49 | 4.42↑0.19 | 14.04↑0.62 | 2.03↑0.03 |
| IM (ours) | 65.66 | 57.47 | 73.24 | 14.57 | 37.48 | 45.29 | 48.95↑2.37 | 4.06↓0.17 | 15.11↑1.69 | 2.44↑0.44 |
| *Alpagasus Dolly 9k (9,229 training examples)* | | | | | | | | | | |
| IT | 64.10 | 56.62 | 69.70 | 7.96 | 32.19 | 42.65 | 45.54 | 4.33 | 21.54 | 2.28 |
| NEFTUNE | 64.20 | 56.69 | 69.51 | 8.99 | 33.91 | 42.62 | 45.99↑0.45 | 4.21↓0.12 | 31.61↑10.07 | 2.84↑0.56 |
| IM (ours) | 64.67 | 55.32 | 74.87 | 12.50 | 36.69 | 43.96 | 48.00↑2.46 | 4.55↑0.22 | 30.77↑9.23 | 2.67↑0.39 |
| *Less Tydiqa (13,533 training examples)* | | | | | | | | | | |
| IT | 64.01 | 56.81 | 64.77 | 12.06 | 36.54 | 55.09 | 48.21 | 4.08 | 5.12 | 1.88 |
| NEFTUNE | 64.03 | 55.09 | 64.02 | 13.84 | 36.65 | 51.21 | 47.47↓0.74 | 4.19↑0.11 | 8.35↑3.23 | 2.58↑0.70 |
| IM (ours) | 64.28 | 56.10 | 65.70 | 17.15 | 34.86 | 54.09 | 48.70↑0.49 | 4.36↑0.28 | 10.10↑4.98 | 2.88↑1.00 |
| *Less MMLU Chat (13,533 training examples)* | | | | | | | | | | |
| IT | 64.74 | 57.42 | 62.94 | 9.53 | 33.13 | 55.35 | 47.18 | 3.86 | 4.42 | 1.20 |
| NEFTUNE | 65.21 | 57.43 | 63.14 | 9.45 | 35.89 | 55.32 | 47.74↑0.56 | 4.06↑0.20 | 6.22↑1.80 | 1.06↓0.14 |
| IM (ours) | 63.95 | 56.34 | 64.76 | 12.52 | 36.94 | 52.55 | 47.84↑0.66 | 4.54↑0.68 | 9.78↑5.36 | 1.93↑0.73 |
| *Less BBH ICL (13,533 training examples)* | | | | | | | | | | |
| IT | 63.83 | 62.04 | 75.92 | 6.90 | 38.93 | 42.07 | 48.28 | 4.78 | 36.20 | 2.36 |
| NEFTUNE | 63.88 | 58.83 | 67.97 | 13.54 | 38.63 | 51.33 | 49.03↑0.75 | 5.05↑0.27 | 39.81↑3.61 | 2.87↑0.51 |
| IM (ours) | 64.14 | 56.72 | 71.12 | 13.56 | 39.03 | 50.34 | 49.15↑0.87 | 5.03↑0.25 | 44.15↑7.95 | 3.56↑1.20 |
| *LIMA (1,030 training examples)* | | | | | | | | | | |
| IT | 63.92 | 58.29 | 71.96 | 16.01 | 39.27 | 43.29 | 48.79 | 4.77 | 33.06 | 2.58 |
| 10 epoch NEFTUNE | 63.66 | 57.67 | 73.03 | 15.95 | 38.77 | 43.14 | 48.70↓0.09 | 4.79↑0.02 | 30.51↓2.55 | 2.43↓0.15 |
| IM (ours) | 64.49 | 58.21 | 75.55 | 17.06 | 38.84 | 43.45 | 49.60↑0.81 | 4.83↑0.06 | 32.94↓0.12 | 2.47↓0.11 |

**Evaluation Benchmarks.** Our study conducts a comprehensive analysis of 21 NLP datasets, focusing on a suite of canonical NLP benchmarks and their capacity for open-ended language generation. For canonical NLP benchmarks, the evaluation is organised into six categories (18 tasks in total): (1) *Language Understanding and Knowledge* includes MMLU [24], PIQA [6], OpenbookQA [41], HellaSwag [74], and LAMBADA [45]; (2) *Multilinguality* contains LAMBADA Multilingual [45], WMT 2014 [8], and WMT 2016 [52]; (3) *Commonsense Reasoning* features Winograd schema challenge (WSC) [35], WinoGrande [50], AI2 Reasoning Challenge (ARC) [16], and CoQA [49]; (4) *Math and Coding Reasoning* includes GSM8K [17], and HumanEval [12]; (5) *Safety and Helpfulness* comprises TruthfulQA [38], ToxiGen [21], and Hendrycks Ethics [23]. (6) *Big Bench Hard (BBH)* dataset [60] is included to assess models. Our models are also tested for their open-ended text generation capabilities using model-based evaluations, specifically through MT-Bench [77], AlpacaEval 1.0 and 2.0 [37], where the AlpacaEval 1.0 compares the model outputs against the text_davinci_003 evaluated by GPT-4 and the AlpacaEval 2.0 compares the model outputs against GPT-4 outputs evaluated by GPT-4 Turbo. See evaluation details in Appendix §B.

**All Comparison Approaches.** In our study, we mainly conduct experiment using the LLAMA-2-7B-BASE and LLAMA-2-13B-BASE [62], and the OPT-6.7B [75] models. We report model performance trained on LLAMA-2-7B-BASE if not specified. We compare with NEFTUNE [31] as the baseline, which adds noise to the embedding during the instruction tuning to increase the robustness of instruction-tuned models. See hyperparameter and implementation details in Appendix §C.

## 4.2 Main Results

In this section, we first evaluate the model performance of our approach and baselines across various tasks. Then we investigate the key factors that contribute to the effectiveness of our approach. Below we will discuss our findings in detail.

**#1: Our approach IM can improve the performance of instruction tuning on various NLP tasks and open-ended generation benchmarks.** Figure 1 provides a summary of the model's performance across both traditional NLP tasks and the AlpacaEval 1.0 benchmark. Table 1 offers

a detailed breakdown of experimental results for traditional NLP tasks across six categories, as well as performance on additional benchmarks for open-ended generation (*i.e.,* MT-Bench and AlpacaEval). The experimental results show that our approach (IM) can improve the performance of instruction tuning on various NLP tasks and open-ended generation benchmarks. Specifically, on the `Alpagasus Dolly 3k` dataset, IM improves the overall mean score of NLP tasks to 48.95, an increase of 2.37 points from the baseline. Similarly, on the `Alpagasus Dolly 9k` dataset, we observe an improvement of 2.46 points in the mean NLP score. These improvements are mirrored in the LLM-based evaluations. Specifically, IM raises scores on the AlpacaEval 1.0 benchmark, achieving approximately a ten-point increase on the `Alpagasus Dolly 9k` dataset and doubling the performance on datasets such as `Less MMLU Chat` and `Less Tydiqa`. However, the extent of improvement varies across different datasets. For example, the `LIMA` dataset shows more modest gains, prompting our further analysis to understand the factors influencing the effectiveness of IM.

**#2: Our approach IM is especially beneficial for datasets characterised by lengthy instructions or prompts paired with comparably brief outputs.** To better understand the impact factors on the effectiveness of IM, we extend our experiments to more instruction-tuning datasets, such as `Science Literature`, `Code Alpaca` and `Tulu V2`. Interestingly, as shown in Figure 2 Left, we find that IM is particularly effective in scenarios where datasets characterised by lengthy instructions and shorter outputs, such as `Less MMLU Chat` and `Less BBH ICL`. For example, in datasets like `Less MMLU Chat` and `Less Tydiqa`, IM shows remarkable efficacy. In contrast, the `Tulu V2` dataset, with an instruction to output length ratio of about 0.5, benefits less compared to the `Science Literature` dataset, which has a much higher ratio of 24.7. We hypothesise that this trend can be attributed to the tendency of language models trained on datasets with shorter outputs to overfit. In cases where the instructions are longer, IM acts as an effective form of regularisation, mitigating this issue. For further details on the experimental setup, refer to the Appendix in §C.

**#3: Our approach IM performs better with fewer training examples.** We find that another important factor in the effectiveness of IM is the quantity of training examples. Specifically, we design additional experiments by sampling different numbers of examples from the `Tulu V2` datasets, which contain about 320k training examples and achieve a modest improvement compared to other datasets in Figure 2 Left. We ensure that our samples maintain an instruction-to-output length ratio of around 10. As all these samples are from `Tulu V2`, we can assume they are from the same distribution. As shown in Figure 2 Right, IM demonstrates substantial performance improvements on the AlpacaEval 1.0 as the number of training examples decreases. This suggests that IM could be particularly valuable for developing robust models in resource-constrained scenarios or under the SAH. For further details on the experimental setup, please refer to the Appendix in §C.

### 4.3 Instruction Modelling Mitigates Overfitting of Instruction Tuning

This section explores the underlying interpretation behind the effectiveness of our approach. Our experimental results demonstrate that IM can alleviate the overfitting problem of Instruction Tuning. Below we will discuss our findings in detail.

**#1. Train and test loss analysis.** Figure 3 clearly illustrates the effectiveness of our approach IM in mitigating overfitting issues compared to IT. For both IM and IT, here we only compute the loss over the output part. In the training loss distribution for the `LIMA` dataset, IM exhibits a slightly higher mean loss of $1.45$ compared to $1.37$ for IT, suggesting that IM does not overfit to the training data as much as IT does. This is further corroborated in the test loss distribution on the `Tulu V2` dataset (using a 10% randomly sampled data set), where IM demonstrates a lower mean test loss of $1.17$ compared to $1.32$ for IT. This indicates that IM maintains better generalisation to new data, emphasising the model's capability to learn effectively without fitting excessively to training examples. For more examples, see Appendix §D.

**#2. BLEU score analysis.** Here we generate outputs using the instructions from the training examples via greedy decoding, and then compare the generated outputs with the ground truth outputs in training examples and report the results. We use the BLEU score (up to n-gram order 4) [46] to measure the similarity between outputs, where a higher score on outputs indicates a higher overlap with training examples. As shown in Table 2, outputs generated by IM consistently have lower BLEU

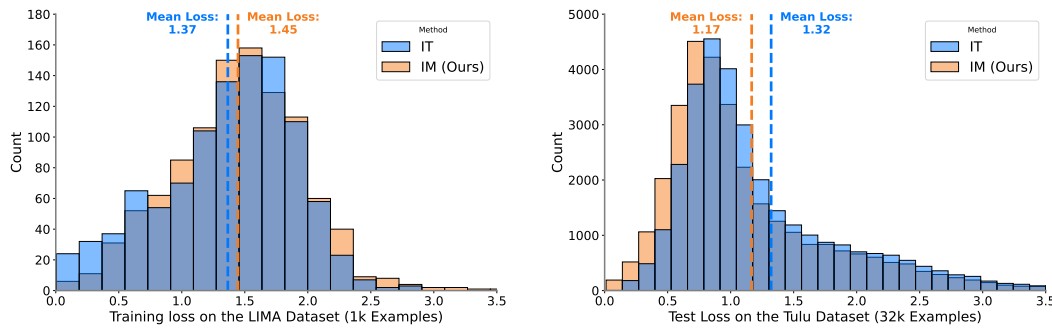

Figure 3: (**Left**) Training loss distribution for each example between our approach INSTRUCTION MODELLING (IM) and INSTRUCTION TUNING (IT) on the LIMA dataset. (**Right**) Test loss distribution for each example between IM and IT on the Tulu V2 dataset, using a 10% randomly sampled data for efficacy. Mean losses are marked by dashed lines. For both IM and IT, here we only compute the loss over the output part. IM has a higher train loss with lower test loss, suggesting that IM effectively mitigates the overfitting issues compared to IT. See Appendix §D for more examples.

Table 2: Average BLEU Score comparison of IM and IT, where a lower score indicates less overfitting. Green and red arrows indicate performance changes against the baseline (IT).

|  | LIMA | Less Tydiqa | Less MMLU Chat | Less BBH ICL | Alpagasus Alpaca 5k | Alpagasus Dolly 9k | Alpagasus Dolly 3k |
|---|---|---|---|---|---|---|---|
| **IT** | 18.15 | 69.21 | 72.43 | 60.96 | 72.26 | 61.76 | 60.99 |
| **IM** (ours) | 17.30↓0.85 | 65.63↓3.58 | 69.20↓3.23 | 53.94↓7.02 | 70.50↓1.76 | 60.61↓1.15 | 59.04↓1.95 |

scores than those generated by IT. This suggests that IM produces outputs have less overlap with the ground truth outputs in training examples, indicating less overfitting.

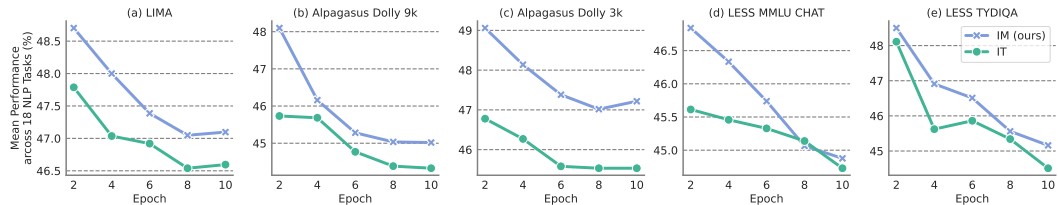

Figure 4: Mean performance on 18 NLP tasks over epochs using LLAMA-2-7B-BASE. This analysis suggests that IM experiences a lower instruction tuning tax compared to IT.

**#3. Instruction Tuning Tax on the NLP tasks.** Previous works show that training LMs with RLHF causes an *Alignment Tax* on the NLP tasks [3, 44]. In this study, we observe that instruction tuning can sometimes lead to diminished model capabilities in some areas, such as multilinguality and commonsense reasoning. To this end, we further explore the impact of instruction tuning on the performance of NLP tasks. Figure 4 illustrates that our approach IM generally has a lower instruction tuning tax compared to IT, suggesting better robustness under low-resource settings. We provide additional experiments for win rates across epochs in Appendix §E.

**#4. Can we simply use KL divergence loss as regularisation for instruction tuning?** In this analysis, we demonstrate that the application of KL divergence loss in instruction tuning, which is widely used as regularisation for aligning LMs [3, 44, 72], cannot easily address the overfitting issue of instruction tuning. Table 3 offers a detailed comparison across various NLP benchmarks and open-ended language generation tasks, particularly using AlpacaEval 2.0, with models trained with and without KL divergence loss. Our findings are as follows: (1) Incorporating KL Loss reduces overfitting and reduces the performance degradation on traditional NLP tasks. For example, on the

Table 3: Performance on 18 NLP benchmarks and AlpacaEval 2.0. Green and red arrows indicate performance changes against the baseline (LLAMA-2-7B-BASE). This analysis suggests that while applying KL Loss in the instruction tuning helps mitigate performance degradation in NLP tasks, it substantially harms the model performance in open-ended generation tasks.

| | | LIMA (1K) | | ALPAGASUS DOLLY (9K) | |
| --- | --- | --- | --- | --- | --- |
| | LLAMA-2-7B-BASE | IT w/o KL Loss | IT w/ KL Loss | IT w/o KL Loss | IT w/ KL Loss |
| **NLP Tasks** | 49.32 | 48.79↓0.53 | 49.26↓0.06 | 45.54↓3.78 | 49.31↓0.01 |
| **AlpacaEval 2.0** | 0.01 | 2.58↑2.57 | 0.06↑0.05 | 2.28↑2.27 | 0.04↑0.03 |

`Dolly` dataset, incorporating KL Divergence Loss leads to less instruction tuning tax in NLP tasks, with scores rising from $45.54$ to $49.31$. (2) KL Loss detrimentally affects the model's instructions following abilities. For example, on the `LIMA` dataset, we observe a substantial decrease in AlpacaEval 2.0 from $2.58$ to $0.06$. For additional experiments and implementation details, see Appendix §F.

## 4.4 Further Analysis

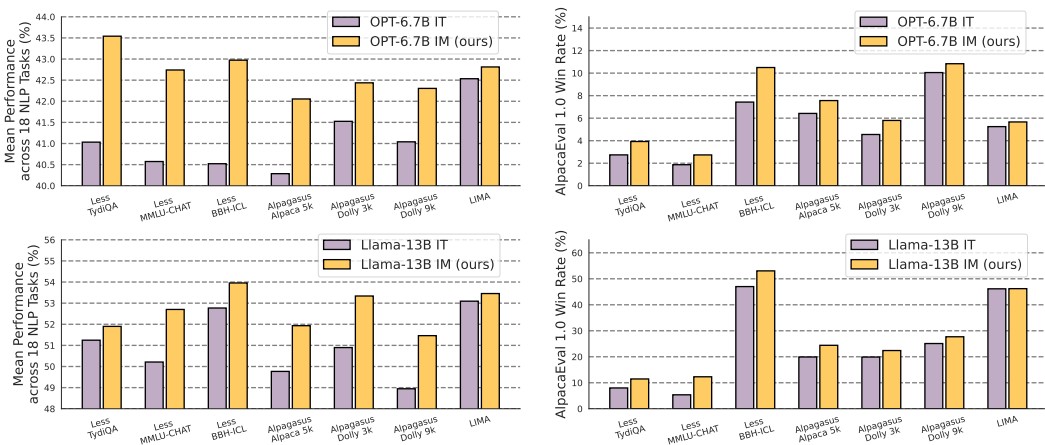

Figure 5: Comparison of INSTRUCTION TUNING (IT) and INSTRUCTION MODELLING (IM) methods using OPT-6.7B (**Top Row**) and LLAMA-2-13B-BASE (**Bottom Row**) trained on 7 instruction tuning datasets. (**Left**) The mean performance across 18 NLP tasks. (**Right**) The win rate on the AlpacaEval 1.0 benchmark.

**#1. The advantage of our proposed method persists with different language models and sizes.** As shown in Figure 5, our analysis demonstrates that our proposed method IM consistently outperforms the IT across different models and sizes, including OPT-6.7B and LLAMA-2-13B-BASE, on 18 traditional NLP tasks and the AlpacaEval 1.0 benchmark. These findings underline the effectiveness of our approach irrespective of the underlying language model or its scale.

**#2. Relationship between the model output length and the win rate.** In this analysis, we explore the potential connection between win rates on the AlpacaEval and the increased output length [37, 76, 31]. As shown in Figure 6, our result reveals that our approach IM does not necessarily generate longer outputs than IT across different data utilisation levels from the `Tulu V2` dataset. Specifically, the output lengths for both approaches are similar despite varying levels of data utilisation. Furthermore, IM consistently outperforms the IT, suggesting that improvements in performance as measured by win rates on the AlpacaEval 1.0 are not dependent on the output length. We provide additional analysis on other instruction tuning datasets under the SAH in Appendix §G.

**#3. Our proposed method IM could further improve the model performance with NEFTUNE.**
Table 4 demonstrates the combined effects of our proposed method IM and NEFTUNE on performance across various NLP tasks and the AlpacaEval 1.0 benchmark. The integration of NEFTUNE with IM

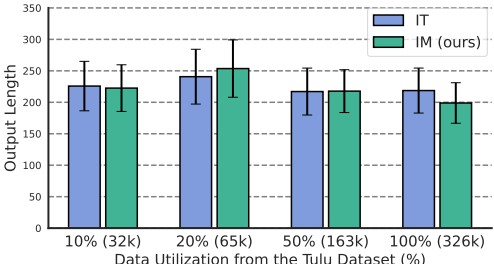 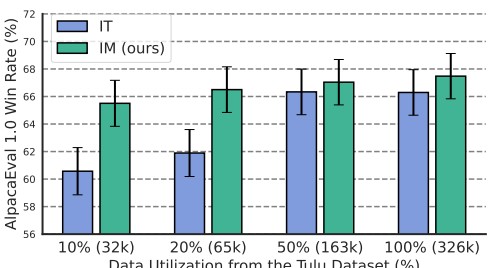

Figure 6: (**Left**) Output length comparison between our approach INSTRUCTION MODELLING (IM) and INSTRUCTION TUNING (IT) across various data utilisation levels from the `Tulu V2` dataset, as evaluated on the AlpacaEval dataset. (**Right**) Performance comparison (measured by win rate) between IM and IT on the AlpacaEval 1.0 across various data utilisation levels from the `Tulu V2` dataset. This analysis suggests that the improvement provided by IM is not necessarily associated with the increased output lengths. See more length analysis in Appendix §G.

Table 4: Performance comparison of IM and IM +NEFTUNE on AlpacaEval 1.0 and various NLP benchmarks. Green and red arrows indicate performance changes against the baseline (IM). This analysis shows that adding NEFTUNE to IM could further improve model performance.

|  | LIMA | Less Tydiqa | Less MMLU Chat | Less BBH ICL | Alpagasus Alpaca 5k | Alpagasus Dolly 9k | Alpagasus Dolly 3k |
|---|---|---|---|---|---|---|---|
| **AlpacaEval 1.0 Win Rate** | | | | | | | |
| **IM** | 32.94 | 10.10 | 9.78 | 44.15 | 19.52 | 30.77 | 15.11 |
| **IM +NEFTUNE** | 30.77↓2.17 | 23.41↑13.31 | 12.45↑2.67 | 48.25↑4.10 | 32.07↑12.55 | 38.28↑7.51 | 23.35↑8.24 |
| **Mean Performance Across 18 NLP Tasks** | | | | | | | |
| **IM** | 49.60 | 48.70 | 47.84 | 49.15 | 47.47 | 48.00 | 48.95 |
| **IM +NEFTUNE** | 49.47↓0.13 | 49.44↑0.74 | 47.73↓0.11 | 48.62↓0.53 | 48.70↑1.23 | 48.63↑0.63 | 49.54↑0.59 |

generally further improves the win rates in AlpacaEval 1.0, showing notable improvements in several datasets such as a 13.31% increase on `Less Tydiqa` and a 12.55% boost on `Alpagasus Alpaca 5k` (in absolute). However, this combination leads to a performance drop in certain contexts, such as a lower performance on NLP tasks on `Less MMLU Chat` and `Less BBH ICL`. This indicates that while NEFTUNE may enhance model robustness under certain conditions, its benefits are context-dependent, highlighting the need for the careful application of NEFTUNE when used in conjunction with IM to optimise effectiveness across diverse evaluation settings.

## 5 Epilogue

**Conclusion.** Our study proposes INSTRUCTION MODELLING, which trains LMs with loss over instructions rather than outputs only. Our experimental evaluations demonstrate that our approach largely improves the performance of LMs on both NLP tasks and open-ended generation benchmarks in some scenarios, especially under the Superficial Alignment Hypothesis and low-resource setting where minimal training data is used for instruction tuning. Our analysis has shed light on two key factors that influence the effectiveness of our approach, (1) the ratio between instruction and output lengths, and (2) the quantity of training data, providing practical insights for optimising instruction-based training methods. Additionally, our analysis reveals the mechanisms behind the effectiveness of IM, particularly its ability to reduce overfitting, showing that applying instruction losses in some scenarios can lead to more robust and adaptable LMs.

**Limitations and Broader Impact.** Here we discuss some potential limitations and the broader impact of our work. Several limitations are outlined as follows: (1) The success of our approach relies on the quality and diversity of the instructions and prompts in the training datasets. Poorly defined or ambiguous instructions may undermine the effectiveness of IM, leading to sub-optimal performance; and (2) It is crucial to ensure that the instructions are ethically sound and free from

harmful or biased content. Training on inappropriate or toxic instructions may result in undesirable outputs. Previous works [5, 7, 4] have extensively discussed the risks and potential harms associated with LMs, including the amplification of undesirable biases learned from training data [5, 2, 9]. Our work has the potential to positively impact the community by helping to mitigate overfitting, resulting in models that are more robust and generalise better to new data, especially in low-resource scenarios. This can enhance the reliability and trustworthiness of AI systems in real-world applications.

## Acknowledgments and Disclosure of Funding

The authors express their gratitude to the NeurIPS reviewers and area chairs for their insightful discussions. Zhengyan is funded by the Research Studentship from University College London (UCL). Bin is funded by the Bloomberg Fellowship. The authors gratefully appreciate the generous support from OpenAI for providing API credits through the OpenAI API Researcher Access Program.

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

## Appendix Overview

The appendix is structured as follows:

**Appendix §A** provides a brief description (with statistical summaries) for instruction tuning datasets.

**Appendix §B** provides details of evaluation benchmarks and settings.

**Appendix §C** provides the experimental setting, implementation details and hyperparameters for all comparison methods used in our experiments.

**Appendix §D** provides the supplementary experimental results to investigate the effect of our approach on training and testing losses.

**Appendix §E** provides the supplementary experimental results to investigate the relationship between the win rate on the AlpacaEval 1.0 and the number of epochs.

**Appendix §F** provides the mathematical formula for the Kullback-Leibler (KL) divergence used in our paper.

**Appendix §G** provides the supplementary experimental results to investigate the relationship between the output length and the number of epochs.

## A  Instruction Tuning Dataset

In this work, we use 13 popular datasets from previous instruction tuning research. For the `WizardLM`, `Sharegpt`, `Science Literature`, and `Code Alpaca` datasets, we directly use the subset provided by the previous work [30]. Refer to the dataset statistics in Table 5. In addition, we provide an analysis of the output length distribution for `LIMA`, `Alpagasus Dolly 3k`, `Alpagasus Dolly 9k`, `Alpagasus Alpaca 5k`, `Less MMLU Chat`, `Less Tydiqa`, and `Less BBH ICL` datasets, as shown in Figure 7.

Table 5: Statistical summary for various instruction tuning datasets. The table includes sample sizes, the average total length of instructions and outputs, the average output length, and the average instruction length with their standard deviations, and ratio calculations.

| Dataset | Size | Total | Output | Output Std | Instruction | Instruction Std | Output/Instruction | Instruction/Output |
|---|---|---|---|---|---|---|---|---|
| LIMA | 1,030 | 484.47 | 442.75 | 491.34 | 41.72 | 79.28 | 10.6124 | 0.0942 |
| Less MMLU Chat | 13,533 | 225.19 | 8.24 | 16.42 | 216.95 | 301.64 | 0.0380 | 26.3316 |
| Less Tydiqa | 13,533 | 172.44 | 25.13 | 42.62 | 147.31 | 235.37 | 0.1706 | 5.862 |
| Less BBH ICL | 13,533 | 262.03 | 61.44 | 92.55 | 200.60 | 196.79 | 0.3063 | 3.265 |
| Alpagasus Dolly 3k | 2,996 | 111.91 | 68.08 | 106.38 | 43.83 | 107.53 | 1.5530 | 0.6439 |
| Alpagasus Dolly 9k | 9,229 | 73.40 | 56.62 | 48.91 | 16.79 | 11.33 | 3.3727 | 0.2965 |
| Alpagasus Alpaca 5k | 5,305 | 48.29 | 30.81 | 34.44 | 17.48 | 12.45 | 1.7631 | 0.5672 |
| Tulu V2 | 326,181 | 541.16 | 343.56 | 575.32 | 197.60 | 345.99 | 1.7387 | 0.5751 |
| Tulu V2 (10%) | 32,618 | 517.45 | 338.96 | 562.74 | 178.49 | 345.72 | 1.8991 | 0.5266 |
| Tulu V2 (50%) | 163,090 | 515.63 | 340.67 | 571.06 | 174.97 | 343.45 | 1.9470 | 0.5136 |
| Tulu V2 (20%) | 65,236 | 504.56 | 336.89 | 562.46 | 167.68 | 331.24 | 2.0092 | 0.4977 |
| WizardLM | 30,000 | 350.05 | 258.35 | 182.98 | 91.71 | 86.09 | 2.8170 | 0.3550 |
| Sharegpt | 50,000 | 1035.39 | 831.15 | 757.10 | 204.24 | 344.51 | 4.0695 | 0.2457 |
| Science Literature | 7,544 | 1196.08 | 46.46 | 57.34 | 1149.62 | 905.99 | 0.0404 | 24.7417 |
| Stanford Alpaca | 52,002 | 63.77 | 45.18 | 44.97 | 18.59 | 12.42 | 2.4302 | 0.4115 |
| Code Alpaca | 20,022 | 49.74 | 27.40 | 27.35 | 22.34 | 10.67 | 1.2262 | 0.8156 |

## B  Evaluation Datasets and Details

We use the open-source repositories, LM-Evaluation Harness[3] and Huggingface Dataset[4] as the evaluation tools. We describe our evaluation setup below:

---

[3] https://github.com/EleutherAI/lm-evaluation-harness
[4] https://huggingface.co/docs/datasets

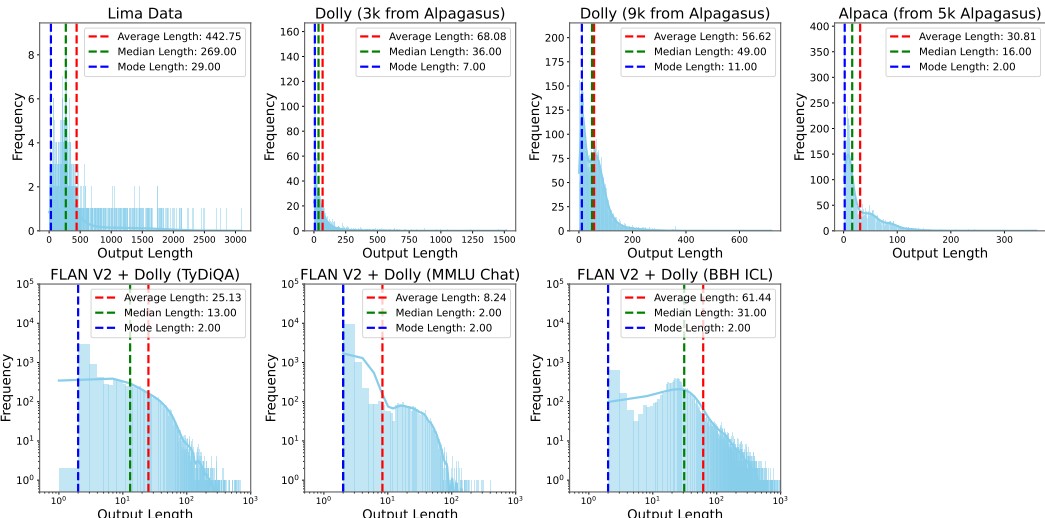

Figure 7: Distribution of output lengths of instruction tuning datasets. This figure presents histograms for the distribution of output lengths across seven datasets, including `LIMA`, `Alpagasus Dolly 3k`, `Alpagasus Dolly 9k`, `Alpagasus Alpaca 5k`, `Less MMLU Chat`, `Less Tydiqa`, and `Less BBH ICL`. Each subplot displays the frequency of output lengths with key statistical indicators: the average (red dashed line), median (green dashed line), and mode (blue dashed line) of each dataset. The last three subplots employ a logarithmic scale on both axes to better illustrate data spread.

**MMLU.** We evaluate the model using the dataset at the huggingface dataset [5]. We follow the protocol outlined in HuggingFace Open LLM Leaderboard [6]. The evaluation uses multiple-choice questions formatted as the question followed by four choices (A, B, C, D) and prompting for an answer. We calculate the mean accuracy (acc) across test examples.

**BBH.** The model evaluation utilises the dataset at the huggingface dataset [7], specifically tested on the 'test' split without the use of few-shot examples. We follow the setup in previous works [30, 60]. The evaluation metric is the exact match score, averaged (mean) to assess performance. Generation is constrained to a maximum of 1024 tokens, with termination upon encountering specific delimiters such as "", "Q", or double newlines. The generation is greedy decoding (temperature set to 0.0) and does not use sampling. Answer extraction employs regex patterns to identify responses immediately following "the answer is" and captures only the first occurrence.

**GSM8K.** We evaluate using the dataset at the huggingface dataset [8], focusing on arithmetic problem-solving in the 'test' split. We follow the HuggingFace Open LLM Leaderboard to 8 few-shot examples. Exact match is the chosen metric, with case insensitivity and select regex-based filtering of common punctuation and formatting characters to ensure precise validation of numerical answers. The primary focus is on extracting and comparing the final numerical answer to the model's output using a strict regex-based match setup.

**HumanEval.** We evaluate using the dataset and the evaluation code from the previous work [30]. We report the performance of the pass@1. We perform the decoding using two different temperatures, 0.1 and 0.7. We report the better pass@1 from these two decoding results.

---

[5] https://huggingface.co/datasets/hails/mmlu_no_train
[6] https://huggingface.co/spaces/HuggingFaceH4/open_llm_leaderboard
[7] https://huggingface.co/datasets/lukaemon/bbh
[8] https://huggingface.co/datasets/gsm8k

**ARC.** The evaluation setup for the dataset at the huggingface dataset [9] utilises a multiple-choice format. We follow the HuggingFace Open LLM Leaderboard to 25 few-shot examples. The performance metric used is mean normalised accuracy (acc_norm).

**CoQA.** We conduct the model evaluation on the dataset at the huggingface dataset [10]. We follow the HuggingFace Open LLM Leaderboard to 0 few-shot examples. The output generation terminates upon encountering a new line followed by "Q:". The mean F1 score is used as the evaluation metric.

**PIQA.** Evaluation on the dataset at the huggingface dataset [11] involves a multiple-choice. The evaluation incorporates 10 few-shot examples, according to the LIMIT [32]. Performance is measured using the mean normalized accuracy (acc_norm).

**OpenBookQA.** The dataset at the huggingface dataset [12] is evaluated in a multiple-choice format. The mean normalized accuracy (acc_norm) is used as the evaluation metric.

**LAMBADA.** The evaluation of the model on the dataset at the huggingface dataset [13] is performed using a loglikelihood output type. The mean accuracy is used as the evaluation metric.

**HellaSwag.** In the 'hellaswag' dataset at the huggingface dataset [14], model evaluation is conducted using a multiple-choice format. We follow the HuggingFace Open LLM Leaderboard to 10 few-shot examples. The mean normalized accuracy (acc_norm) is used as the evaluation metric.

**The Winograd Schema Challenge.** The evaluation is conducted using a multiple-choice format on the 'test' split at the huggingface dataset [15]. The mean accuracy is used as the evaluation metric.

**Winogrande.** The 'winogrande' dataset is assessed using a multiple-choice format at the huggingface dataset [16]. We follow the HuggingFace Open LLM Leaderboard to 5 few-shot examples. The mean accuracy is used as the evaluation metric.

**LAMBADA.** For this dataset, evaluation is conducted using the loglikelihood output type on the 'test' split at the huggingface dataset [17]. This variant focuses on predicting the last word of text passages in English. The mean accuracy is used as the evaluation metric.

**Translation Benchmarks WMT.** The evaluation of the translation capabilities is performed on the WMT 2014[18] and WMT 2016[19] datasets at the huggingface dataset. Here we use the 'ter' score as the evaluation metric.

**TruthfulQA.** We use the dataset at the huggingface dataset [20]. We follow the setup at the HuggingFace Open LLM Leaderboard using the 6 few-shot examples. The mean accuracy is used as the evaluation metric.

**ToxiGen.** We use the dataset at the huggingface dataset [21]. The task is assessed using a multiple-choice framework to evaluate the model's capability to identify hateful content in text statements. The mean accuracy is used as the evaluation metric.

---

[9] https://huggingface.co/datasets/allenai/ai2_arc
[10] https://huggingface.co/datasets/EleutherAI/coqa
[11] https://huggingface.co/datasets/piqa
[12] https://huggingface.co/datasets/openbookqa
[13] https://huggingface.co/datasets/lambada
[14] https://huggingface.co/datasets/hellaswag
[15] https://huggingface.co/datasets/winograd_wsc
[16] https://huggingface.co/datasets/winogrande
[17] https://huggingface.co/datasets/EleutherAI/lambada_openai
[18] https://huggingface.co/datasets/wmt14
[19] https://huggingface.co/datasets/wmt16
[20] https://huggingface.co/datasets/truthful_qa
[21] https://huggingface.co/datasets/skg/toxigen-data

**Hendrycks Ethics.** We use the dataset at the huggingface dataset [22], with a multiple-choice format. The model aims to detect whether described actions in various contexts are ethically wrong. The prompt format integrates a specific scenario followed by a structured question: "Is this wrong?" and then prompts for an answer with options 'no' or 'yes'. The mean accuracy is used as the evaluation metric.

## C  Implementation Details

**Experimental Design for Figure 2 Left.** Here we present a detailed experimental design for Figure 2 Left. We perform experiments on a variety of datasets, including `LIMA`, `Alpagasus Dolly 3k`, `Alpagasus Dolly 9k`, `Alpagasus Alpaca 5k`, `Less MMLU Chat`, `Less Tydiqa`, `Less BBH ICL`, `Tulu V2`, `Code Alpaca`, `Stanford Alpaca`, `Science Literature`, `WizardLM`, and `Sharegpt`. Furthermore, to evaluate the effectiveness of IM on datasets with different instruction-to-output length ratios, we select three subsets from `Tulu V2`. Each subset contains 3,000 training examples, with instruction-to-output length ratios of approximately 5, 10, and 15, respectively.

**Experimental Design for Figure 2 Right.** Here we provide a detailed experimental design for Figure 2 Right. We strategically sampled varying sizes of training examples from the `Tulu V2` dataset to investigate the effectiveness of IM with different sizes of training examples. Starting with approximately 320,000 examples in the `Tulu V2` dataset, we create subsets of data ranging from as few as 1,000 to as many as 35,000 examples. These subsets were selected randomly, ensuring a representative mix across different scales. We adhered to a fixed instruction-to-output length ratio of approximately 10 to maintain consistency in training conditions across all samples. We train the LLAMA-2-7B-BASE on all these subsets and evaluate them respectively.

Table 6: Hyperparameters and configurations for supervised fine-tuning.

| Hyperparameter | Assignment |
| --- | --- |
| GPUs | 2 or 4 A100 80G GPUs, 2 48G A6000 GPUs |
| Batch size per GPU | 1 |
| Total batch size | 128 |
| Number of epochs | 2, 3, or 10 |
| Maximum sequence length | 2048 |
| Learning rate | $2 \times 10^{-5}$ |
| Learning rate optimizer | AdamW |
| Adam epsilon | 1e-6 |
| Adam beta weights | 0.9, 0.98 |
| Learning rate scheduler | Linear with warmup |
| Warmup proportion | 0.03 |
| Weight decay | 0 |
| Mixed precision | bf16 |
| Gradient accumulation steps | Calculated dynamically |

**Implementation Details.** In our study, we fine-tune the LLaMA-2-7B, LLaMA-2-13B and OPT-6.7 model using four A100 80G GPUs, with a per-GPU batch size of 1 and a total batch size of 128, employing a learning rate of 2e-5. Training typically proceeds for 2 epochs with a maximum sequence length of 2048 tokens. We utilise gradient accumulation, calculated to effectively distribute training steps across the available hardware, resulting in larger batch sizes despite hardware limitations. We employ mixed precision (bf16), linear learning rate scheduling with a warm-up ratio of 0.03, and a weight decay of 0. To optimise our training, we use DeepSpeed with a stage 3 configuration without offloading. Our setup also includes the usage of Flash Attention [19] and slow tokenization

---

[22]https://huggingface.co/datasets/EleutherAI/hendrycks_ethics

to enhance training efficiency and compatibility. Our code is implemented using Open-Instruct[23], Pytorch[24] and Huggingface[25]. Table 6 lists the hyperparameters.

# D   Train and Test Loss

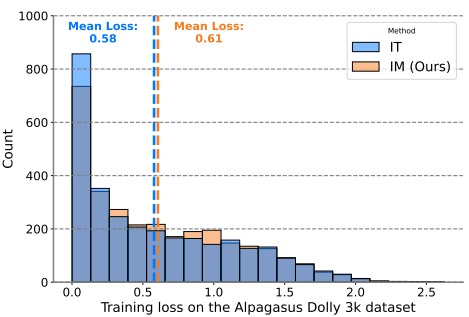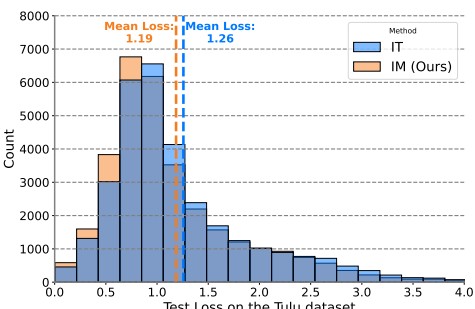

Figure 8:  (**Left**) Training loss distribution for each example between our approach INSTRUCTION MODELLING (IM) and INSTRUCTION TUNING (IT) on the `Alpagasus Dolly 3k` dataset. (**Right**) Test loss distribution for each example between IM and IT on the `Tulu V2` dataset, using a 10% sampled data. Mean losses are marked by dashed lines. For both IM and IT, here we only compute the loss over the output part.  IM has a higher train loss with lower test loss, suggesting that IM effectively mitigates the overfitting issues compared to IT.

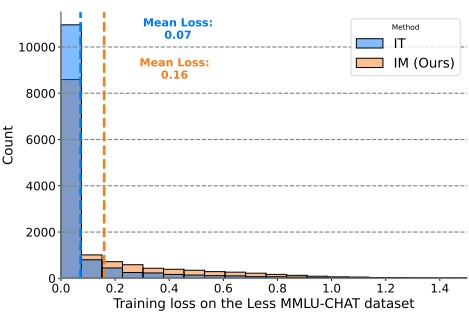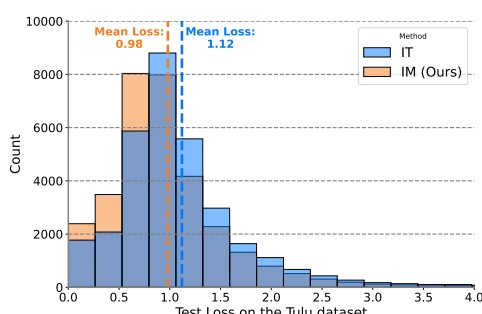

Figure 9:  (**Left**) Training loss distribution for each example between our approach INSTRUCTION MODELLING (IM) and INSTRUCTION TUNING (IT) on the `Less MMLU Chat` dataset. (**Right**) Test loss distribution for each example between IM and IT on the `Tulu V2` dataset, using a 10% sampled data. Mean losses are marked by dashed lines. For both IM and IT, here we only compute the loss over the output part. IM has a higher train loss with lower test loss, suggesting that IM effectively mitigates the overfitting issues compared to IT.

In this section, we provide additional experiments regarding training and testing loss distributions. Figure 8 focuses on the `Alpagasus Dolly 3k` and `Tulu V2` datasets, displaying how IM tends to exhibit higher training losses yet achieves lower test losses compared to IT. Similarly, Figure 9 compares these methods on the `Less MMLU Chat` and `Tulu V2` datasets under analogous conditions.

# E   The impact of Epochs on the Win Rate

Figure 10 illustrates the comparative analysis of AlpacaEval 1.0 scores across different epochs for two datasets, `LIMA` and `Alpagasus Dolly 9k` datasets. We evaluate the performance of IM and IT

---

[23]https://github.com/allenai/open-instruct
[24]https://pytorch.org/
[25]https://huggingface.co/

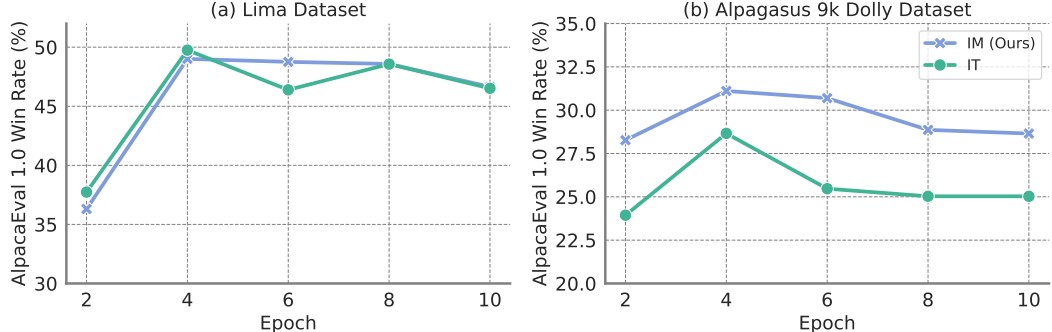

Figure 10: AlpacaEval 1.0 performance trends for IM and IT approaches on the `LIMA` and `Alpagasus Dolly 9k` datasets across different epochs.

over different numbers of epochs. IM consistently surpasses IT in performance on the `Alpagasus Dolly 9k` dataset, while the performance of both approaches is comparable on the `LIMA` dataset.

## F    Applying KL Divergence Loss for Instruction Tuning

In this section, we first briefly introduce the Kullback-Leibler (KL) divergence and then introduce the experimental details. Future work can investigate the effectiveness of parameter-efficient fine-tuning approaches as the regularisation [80, 55, 28].

**Kullback-Leibler Divergence.**    Kullback-Leibler (KL) divergence is commonly employed as a regularisation method in the fine-tuning of LMs, helping to mitigate overfitting by constraining the fine-tuned model to remain similar to the pre-trained model [44]. Specifically, the KL divergence is added to the fine-tuning objective as a per-token regularisation term between the fine-tuned LM $\pi_\theta(x)$, and the pre-trained LM, $\pi^{\text{pre}}(x)$. For supervised fine-tuning with the next token prediction loss, the training objective incorporating KL divergence is computed as follows:

$$\mathcal{L}_{\text{KL}}(\theta) = \mathbb{E}_{x \sim \mathcal{D}} \Big[ \sum_t - \log \pi_\theta(x_t|x_{0:t-1}) + \lambda \sum_t \text{KL}(\pi_\theta(x_t|x_{0:t-1}) || \pi^{\text{pre}}(x_t|x_{0:t-1})) \Big], \quad (5)$$

where $\lambda$ is a regularisation parameter that balances the loss due to the next token prediction and the KL divergence, and $\pi(x_t|x_{0:t-1})$ represents the next token distribution of the fine-tuned or pre-trained LM conditioned on the preceding context.

Table 7: Performance on 18 NLP tasks and AlpacaEval 2.0, with various values of $\lambda$, trained on the (LLAMA-2-7B-BASE).

|  | NLP Tasks | AlpacaEval 2.0 |
|---|---|---|
| LLAMA-2-7B-BASE | 49.32 | 0.01 |
| $\lambda = 0.01$ | 48.81 | 2.58 |
| $\lambda = 0.1$ | 48.77 | 2.44 |
| $\lambda = 1.0$ | 49.26 | 0.06 |

**Ablation study on the effect of $\lambda$.**    In Table 3, we set the value of the $\lambda$ as 1.0. Here we provide additional experiments with different values of $\lambda$. Table 7 presents the model performance on the NLP tasks and AlpacaEval 2.0. This aligns our observations in §4.3.

## G    The impact of Epochs on Output Lengths

Figure 11 illustrates the average output length of various models across different epochs. We report the output length on four different datasets, including `Alpagasus Dolly 3k`, `Alpagasus Dolly`

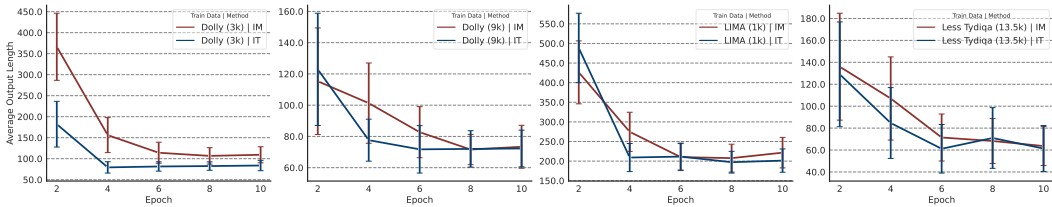

Figure 11: Comparative analysis of output lengths for IM and IT across different epochs on `Alpagasus Dolly 3k`, `Alpagasus Dolly 9k`, `LIMA`, and `Less Tydiqa` datasets.

`9k`, `LIMA`, and `Less Tydiqa`. Each line represents the average output length of a model, with epochs ranging from 2 to 10, and is accompanied by error bars that denote the normalised standard deviation (10%) of the output lengths. Our experimental results show that our approach IM does not consistently increase the output length and that win rates are not necessarily associated with the length of the output.

