# OpenReview forum: "Instruction Tuning With Loss Over Instructions"
_NeurIPS.cc/2024/Conference — NeurIPS 2024 poster_

### Official Review · Reviewer_vkoa · 2024-07-13

**Soundness:** 3
**Presentation:** 3
**Contribution:** 3
**Rating:** 6
**Confidence:** 5

**Summary:**

This paper proposes a new method called Instruction Modelling (IM) for training language models, which applies a loss function to both the instruction and output parts of training data, rather than just the output. Through experiments on diverse benchmarks, the authors show that IM can improve model performance on both NLP tasks and open-ended generation benchmarks compared to standard Instruction Tuning (IT). The effectiveness of IM is influenced by two key factors: the ratio between instruction length and output length in the training data, and the number of training examples. IM is particularly beneficial for datasets with long instructions paired with brief outputs, or when using a small amount of training data. The authors hypothesize that IM's improvements stem from reducing overfitting during instruction tuning.

**Strengths:**

- The paper revisits the fundamental approach to instruction tuning, which typically involves calculating loss only on the output portion of the data. By proposing Instruction Modelling (IM), which applies loss to both the instruction and output parts, the authors challenge this standard practice. This fresh perspective on a widely-used technique is a key strength of the paper.
- The authors conduct extensive experiments across many diverse benchmarks, demonstrating the broad applicability and effectiveness of their proposed method.
- The paper identifies and analyzes two crucial factors influencing IM's effectiveness: the ratio between instruction length and output length in training data, and the number of training examples. This analysis provides valuable insights for practitioners on when and how to best apply the IM approach, particularly in low-resource scenarios.

**Weaknesses:**

- Similar ideas and conclusions have been proposed by previous work [1].
- While the paper presents empirical results showing the effectiveness of Instruction Modelling (IM), it lacks a strong foundation explaining why applying loss to instructions works. The authors hypothesize that IM reduces overfitting, but a more rigorous theoretical analysis could provide deeper insights into the mechanism behind IM's success.
- The experiments primarily use LLaMA-2 and OPT models. While these are significant models, the paper doesn't explore how IM performs across a wider range of model architectures or sizes, e.g., whether the conclusion also holds for a 34B or a 70B model.

[1] Instruction Fine-Tuning: Does Prompt Loss Matter? https://arxiv.org/pdf/2401.13586v2

**Questions:**

- Are there any possible experiments that you can do to further explain why applying loss to instruction works?
- Do you think when we have a significantly larger size of instruction tuning data, the conclusion still holds? For example, we see the recent release of Llama-3 models, which adopted 10+ million instructions. And which factor is more important: the ratio between output and the instruction length or the number of instruction tuning samples?

**Limitations:**

Yes. But the authors do not have a mandatory paper checklist.

---

> ### Author Rebuttal · Authors · 2024-08-01
>
> We sincerely thank the reviewer for the thoughtful and constructive feedback. We are grateful for the reviewer's positive feedback: the fresh perspective, the extensive experiments, and the valuable analysis of crucial factors influencing IM's effectiveness. We would like to address the reviewer's valuable feedback as follows:
>
> > Limitations: But the authors do not have a mandatory paper checklist.
>
> We have the checklist from the Page 23 to 29 in our main PDF file. We recognize that it may not have been easily noticeable, and we will ensure that in the revised version.
>
> > Similar ideas and conclusions have been proposed by previous work \[1\].
>
> We appreciate the reviewer bringing attention to \[1\]. As noted in our related work section, we have already discussed this paper. While \[1\] introduces a hyperparameter to control the degree of instruction loss during training and investigates its impact, our work proposes a broader guideline. Instead of introducing new hyperparameters, we focus on when and how to include loss over instruction effectively and explore the underlying mechanisms. Our approach provides a more practical and widely applicable framework.
>
> > While the paper presents empirical results showing the effectiveness of Instruction Modelling (IM), it lacks a strong foundation explaining why applying loss to instructions works. The authors hypothesize that IM reduces overfitting, but a more rigorous theoretical analysis could provide deeper insights into the mechanism behind IM's success.
>
> We appreciate the reviewer's feedback. While we acknowledge the importance of rigorous theoretical analysis, it falls beyond the scope of our current work. Our paper aims to provide general, empirical insights for instruction tuning. Our key message is that the efficacy of masking user prompts during instruction tuning is an empirical question. We find that including prompt loss during training can be particularly advantageous when the number of instruction tuning data is limited and completions are short.
>
> We recognise the challenges in rigorously proving improvements, especially given the varied knowledge carried by different pre-training models and instruction tuning datasets. A comprehensive theoretical analysis would need to account for these complex, interacting factors. Our empirical findings provide insights for future theoretical investigations into instruction tuning.
>
> > The experiments primarily use LLaMA-2 and OPT models. While these are significant models, the paper doesn't explore how IM performs across a wider range of model architectures or sizes.
>
> We conducted additional experiments on additional models including Phi and Gemma, which show qualitatively similar results (see Table 2 in the rebuttal pdf).  However, we were not able to finetune 60-70b parameter due to computational restrictions.  We will add a note on this point to the limitations section.
>
> > Are there any possible experiments that you can do to further explain why applying loss to instruction works?
>
> We appreciate the reviewer's insightful question. While our current study provides empirical evidence for reducing overfitting through loss and BLEU analysis, we agree that further analyses could offer deeper insights into the underlying mechanisms. Several factors may affect the model's effectiveness, including assessing whether IM improves the factual correctness of responses or reduces toxic content in model outputs, as well as testing if IM enhances robustness to out-of-distribution (OOD) instructions.
>
> To address this aspect, we conducted additional experiments comparing high-quality datasets \[2, 3\] with randomly selected datasets of the same size from the same source datasets. As shown in Table 1 of our attached PDF file, our results indicate that our approach generally performs better even when trained on randomly selected datasets. This finding suggests that our approach is robust across various data qualities and may be particularly beneficial when working with diverse, non-curated datasets. We will include these results in our revised paper.
>
> > Do you think when we have a significantly larger size of instruction tuning data, the conclusion still holds? For example, we see the recent release of Llama-3 models, which adopted 10+ million instructions. And which factor is more important: the ratio between output and the instruction length or the number of instruction tuning samples?
>
> We thank the reviewer for raising these important questions. We would like to clarify that our approach is not intended to replace instruction tuning in all scenarios; rather, we propose that the decision to mask user prompts during this process should be empirically driven. In scenarios with limited instruction tuning data and short completions, incorporating prompt loss during training can be particularly beneficial.
>
> Regarding the relative importance of factors, we view the ratio between output and instruction length and the number of instruction tuning samples as fundamentally equivalent, both representing constraints on instruction tuning resources. Our research demonstrates that exposing the model to additional loss signals through instruction modelling—specifically by including loss on the prompt during training—can lead to more robust and effective models. This approach maximizes the utility of limited resources, potentially enhancing model performance across various tasks.
>
> ### Reference:
> \[1\] Instruction Fine-Tuning: Does Prompt Loss Matter? Arxiv 2024\.
>
> \[2\] LESS: Selecting Influential Data for Targeted Instruction Tuning, ICML 2024
>
> \[3\] AlpaGasus: Training A Better Alpaca with Fewer Data. ICLR 2024\.

---

### Official Review · Reviewer_kinw · 2024-07-13

**Soundness:** 3
**Presentation:** 2
**Contribution:** 3
**Rating:** 7
**Confidence:** 4

**Summary:**

The propose that when updating models using instruction tuning, the models should also be updated based on loss on the instruction itself.

This is a simple change that is un-intuitive, so the successful results are impressive.

**Strengths:**

They include experiments from many different datasets and look at multiple different LLMs. And there results generally hold across them.

**Weaknesses:**

It is unclear what the paper considers an "instruction", there should be some examples of it. At one point it says that "static parts" of the instruction, like a "<user>:" token, are masked from the lost. Does that mean the rest of the instruction is all examples? An instruction like "Tell me the sentiment of this text" is going to be static across examples.

When looking at table 1, although their IM methods wins in terms of average performance, there are many datasets where it performs worse. This is glossed over in their prose.

Their "Loss analysis" experiments are not convincing as they don't fully capture the changes to the model that occur in each setting. By only checking the loss on the continuations, they don't capture how their IM model might be more overfit to the instructions themselves. For example, if you looked that the train loss for the whole Instruction + continuation for each models, the IM model would mostly likely be far lower than the loss for the instruction tuned model. It makes sense that the IM model is going to be worse on the continuations after the same amount of training as updates to get better as the instruction itself will cause conflicts. Discarding the models performance on the Instructions creates a self-fullfilling prophecy of the loss being higher when only the continuations are considered.

**Questions:**

What is considered an "instruction" when training? The text mentions that "static" tokens like "<user>:" are masked out, but many instructions are static, i.e. "Tell me if the second sentence entails the first". If instructions are non-static are they mostly example like in In-Context-Learning?

They mention that IM is most effective in settings where the instructions are long. Is it possible that most of this gain is for essentially continuing pre-training on in-domain data a la https://arxiv.org/abs/2004.10964

Similarly, do you think the success in the SAH settings is possibly an artifact of getting a model that is just trained on a lot more data? How many more tokens does an IM model see than in IT model in that setting? Is there a way to hold the number of tokens seen constant? For example by training the IM model on even fewer examples to show that the gain really is from the training on the instruction?

**Limitations:**

yes

---

> ### Author Rebuttal · Authors · 2024-08-02
>
> We sincerely thank the reviewer for the thoughtful and constructive feedback. We are pleased to receive positive feedback on the extensive experiments and the consistency of our results across diverse settings. We would like to address the reviewer's valuable feedback as follows:
>
> > It is unclear what the paper considers an "instruction", there should be some examples of it. At one point it says that "static parts" of the instruction, like a "<user>:" token, are masked from the lost. Does that mean the rest of the instruction is all examples? An instruction like "Tell me the sentiment of this text" is going to be static across examples.
>
> In our work, the instructions contain the following parts:
> - Static templates. These are pre-defined templates randomly selected (please refer to Lines 4-38 in our code repository file `src/instruction_encode_templates.py` (Please find the link in our abstract).
> - Dynamic instructions from the user. These are the actual user-provided instructions contained in the “instruction” key of the examples.
> - Formatting tokens. We consistently mask out formatting tokens like "\<user\>:" to concentrate on the content.
> ```json
> {
>   "instruction": "In 1994 , the Polish ambassador to South Africa , Mr Durrant , presented the `` Warsaw Cross of Insurrection to SCieniuch 's widow .\nIn 1994 , the Polish ambassador to South Africa , Mr Scieniuch , presented the `` Warsaw Cross of Insurrection to Durrant 's widow .\n\nAre these two sentences paraphrases of each other?\nOptions are:\n 1). no.\n 2). yes.\n",
>   "completion": "1)",
> },
> {
>   "instruction": "Give three tips for staying healthy.\n\n",
>   "completion": "1.Eat a balanced diet and make sure to include plenty of fruits and vegetables. \n2. Exercise regularly to keep your body active and strong. \n3. Get enough sleep and maintain a consistent sleep schedule.",
> }
> ```
> We will add these examples in the revised paper.
>
> > When looking at table 1, although their IM methods wins in terms of average performance, there are many datasets where it performs worse. This is glossed over in their prose.
>
> We will modify the text to acknowledge this point. We also respectfully present a different perspective: it is not trivial to discern a clear trend or pattern of how different methods affect model performance through a single or small number of tasks. This is precisely why our work includes 23 NLP benchmarks from 6 different categories and 3 LLM-based evaluations. Our analysis aims to provide a holistic view of how our approach and baselines impact downstream task performance. The observed performance variations across datasets offer valuable insights for future research.
>
> > Their "Loss analysis" experiments are not convincing as they don't fully capture the changes to the model that occur in each setting.
>
> Our loss analysis focuses on the loss of continuations, as including the loss over the instruction part for our approach could lead to an unfair comparison. In our paper, we also conducted a BLEU Score analysis to investigate the potential overfitting issue. This analysis compares the overlap between model outputs and the ground truth outputs in training examples. Our results show that our approach produces outputs with less overlap with the ground truth outputs in training examples, indicating less overfitting compared to the baseline models.
>
> > They mention that IM is most effective in settings where the instructions are long. Is it possible that most of this gain is for essentially continuing pre-training on in-domain data a la https://arxiv.org/abs/2004.10964
>
> We thank the reviewer for this insightful question. Our approach differs from continued pre-training in several aspects:
> - Our approach exposes the model to more loss signals under limited resources, whereas continued pre-training leverages **large-scale** unlabeled text to learn in-domain knowledge.
> - Our method does not require an additional source of data and can be applied almost for free within existing finetuning pipelines, and we can therefore expect to see wide use.
> - Continued pre-training typically targets performance improvement in specific domains or tasks. In contrast, our approach demonstrates effectiveness across a broad spectrum of 23 NLP benchmarks and 3 LLM-based evaluations, indicating greater generalizability.
> - Our method specifically targets the relationship between instructions and their corresponding outputs, a focus not present in traditional continued pre-training approaches.
>
> We will clarify these differences in our revised version.
>
> > Similarly, do you think the success in the SAH settings is possibly an artifact of getting a model that is just trained on a lot more data? How many more tokens does an IM model see than in IT model in that setting? Is there a way to hold the number of tokens seen constant? For example by training the IM model on even fewer examples to show that the gain really is from the training on the instruction?
>
> We would like to argue that the increased exposure to tokens in our approach is not a limitation, but rather a key advantage of our method. This aligns with observations in other areas of LLM research. For instance, Yi Tay discussed in his blog [1] that one potential reason decoder models outperform encoder-only models is due to greater "loss exposure" in the next token prediction objective compared to the denoising objective.
>
> Similarly, in our approach, exposing the model to more loss signals through instruction modelling is beneficial. This is particularly useful in scenarios with limited instruction tuning resources. By including loss on the prompt during training, we leverage more of the available data, potentially leading to more robust and effective models.
>
> ### Reference:
> [1] https://www.yitay.net/blog/model-architecture-blogpost-encoders-prefixlm-denoising

---

### Official Review · Reviewer_b1op · 2024-07-14

**Soundness:** 3
**Presentation:** 4
**Contribution:** 3
**Rating:** 7
**Confidence:** 4

**Summary:**

This paper proposes Instruction Modeling (IM), which trains LMs by applying a loss function to the instruction and prompt part rather than solely to the output part. The method is found to be effective on NLP tasks and open-ended generation benchmarks. This paper found two key factors that influence the effectiveness of the approach: (1) the ratio between instruction and output lengths, and (2) the quantity of training data. There are also additional analysis that shows IM can reduce overfitting.

**Strengths:**

Generally the paper is well written and covers a wide range of experiments. It forms a comprehensive study on whether insturction tuning should calculate loss over the instruction part. Actually, it's a little surprise to me that calculating loss over the instruction part can improve the performance of the model, and the reasoning behind this explained by the paper is insightful and interesting. Overall I think:

1. The finding is interesting and novel. This is in contrast to the common practice of calculating loss over the output part only. I think it's going to have broad impact on how people finetune LMs in the future if all the claims are true.

2. The experiments and evaluations are comprehensive. THe paper covers a wide range of instruction datasets and quite a few widely used benchmarks.

3. The paper has a lot of details, in both the main body and the appendix. The appendix is very detailed and informative, which is good for reproducibility.

**Weaknesses:**

A part I feel that's missing in the paper is analysis on the **quality of the instruction part**. One reason that people didn't calculate loss over the instruction part is that the instruction part is usually noisy and not well-structured, e.g, ShareGPT has a lot of user shared low-quality instructions. The paper should analyze how the quality of the instruction part affects the performance of the model. Intuitively, if the model is also learning from the noisy instruction part, it's possible that the model will learn undesirable patterns from the instruction part, but I don't see this being discussed in the paper.

Moreover, the number of LMs being finetuned and tested is relatively small. It will be better if more families of LMs and larger models (eg, 60-70B params) are tested.

**Questions:**

1. Do you have any analysis on the quality of the instruction part? How does the quality of the instruction part affect the performance of the model? See "Weaknesses" part.

2. I might have missed this one in the paper, but how do you pack multiple samples during finetuning? Do you concatenate samples to form a fixed length and then trucate them? Or do you add padding tokens?

3. Do you have any scenarios where the IM is significantly worse than the baseline? If so, can you provide some examples? I want to know under what conditions IM is not effective and should be avoided.

4. Do you think IM can be used as a drop-in replacement for the current finetuning process? Or do you think it should be used in conjunction with the current finetuning process?

**Limitations:**

Limitations are being discussed in the paper.

---

> ### Author Rebuttal · Authors · 2024-08-01
>
> We appreciate the effort and time of the reviewer (b1op). We are grateful for the positive feedback on our paper's comprehensiveness, novelty, potential impact, extensive experiments, and detailed presentation. We are particularly pleased that our well-written reasoning and thorough appendix were recognised. We would like to address the reviewer's valuable feedback as follows:
>
> > A part I feel that's missing in the paper is analysis on the quality of the instruction part.
>
> We thank the reviewer for this insightful comment. In response to this feedback, we have conducted additional experiments to address this aspect of our work.
>
> We acknowledge that it is not trivial to define what constitutes high-quality data, and it is particularly challenging to judge the quality of instructions objectively. Given these difficulties, we have chosen to follow Alpagasus \[1\] in ICLR 2024 and LESS \[2\] in ICML 2024 where the overall quality of instructions and completions has been evaluated.
>
> Specifically, we compare high-quality data subsets identified in previous work \[1,2\] with randomly selected data subsets of the same size from the same underlying source datasets. We utilised four datasets:
>
> - Less Tydiqa (13 533 examples), selected from the source datasets, Flan V2 and Dolly
> - Alpagasus Dolly 3k (2 996 examples), selected from the source dataset Dolly
> - Alpagasus Dolly 9k (9 229 examples), selected from the source dataset Dolly
> - Alpagasus Alpaca 5k (5 305 examples), selected from the source dataset Alpaca
>
> As shown in Table 1 in our attached PDF file, our results show that our Instruction Modeling (IM) approach performs better than baselines on both high-quality and randomly selected datasets. This finding suggests that IM is robust across various data qualities. We will include these results in our revised paper.
>
> > Moreover, the number of LMs being finetuned and tested is relatively small. It will be better if more families of LMs and larger models (eg, 60-70B params) are tested.
>
> We conducted additional experiments on additional model families including Phi and Gemma, which show qualitatively similar results (see Table 2 in the rebuttal pdf).  However, we were not able to finetune 60-70b parameter due to computational restrictions.  We will add a note on this point to the limitations section.
>
> > I might have missed this one in the paper, but how do you pack multiple samples during finetuning? Do you concatenate samples to form a fixed length and then truncate them? Or do you add padding tokens?
>
> We appreciate the opportunity to clarify this point. We follow the standard training paradigm of instruction tuning, where we do not pack training examples. Instead, we add the padding token to the maximum sequence length for each training example. We will make this clearer in our revised paper.
>
> > Do you have any scenarios where the IM is significantly worse than the baseline? If so, can you provide some examples? I want to know under what conditions IM is not effective and should be avoided.
>
> We appreciate the reviewer's interest in the potential limitations of our method. In our extensive experiments, we have not observed scenarios where IM is significantly worse than the baseline. We have observed that when two specific scenarios are not met—namely, (1) limited instruction tuning data and (2) short completions—it generally does not significantly impact performance whether the prompt loss is included during training, leaving the performance more empirical. We acknowledge that there might be specific edge cases or scenarios not covered in our current experiments. For example, if instruction parts in training examples include more toxic or harmful content, understanding how our approach affects model performance in such situations remains an area for future research.
>
> > Do you think IM can be used as a drop-in replacement for the current finetuning process? Or do you think it should be used in conjunction with the current finetuning process?
>
> We thank the reviewer for this question as it allows us to clarify our position. We are not proposing IM as a complete replacement for current fine-tuning processes. Rather, our key message is that whether masking user prompts during instruction tuning could be more empirical. We find that the effectiveness of masking user prompts can vary depending on factors such as the amount of instruction tuning data available and the length of completions. In some cases, including the prompt loss during training might be advantageous. We appreciate the opportunity to clarify this point and will ensure it is clearly stated in our revised paper.
>
> ### Reference:
>
> \[1\] AlpaGasus: Training A Better Alpaca with Fewer Data. ICLR 2024\.
>
> \[2\] LESS: Selecting Influential Data for Targeted Instruction Tuning, ICML 2024

---

### Official Review · Reviewer_z7Cy · 2024-07-14

**Soundness:** 4
**Presentation:** 4
**Contribution:** 4
**Rating:** 7
**Confidence:** 4

**Summary:**

In this work, authors propose to use instruction modeling (using loss over the full instruction-output pair) instead of just instruction tuning (using loss over the output given the instruction) as a method for supervised finetuning on LLMs. The authors demonstrate consistent gains over multiple benchmarks using this simple technique and hypothesize that the gains occur due to reduced overfitting to instruction tuning dataset.

**Strengths:**

1. The proposed method is quite simple and scalable as such would be of great interest to practitioners in the community.
2. The authors try to characterize the gains and attribute it to the extra signal obtained in the case of short outputs as well as to reduced overfitting of the approach. Both characterizations seem intuitive in explaining the results obtained. Further, the authors show that proposed method is complementary to NEFTUNE, another recent method which results in consistent empirical gains.

**Weaknesses:**

1. Limited Models Explored: While the authors do a good job of experimenting with multiple instruction datasets, the experiments are only done with LLama-2 (7B and 13B) and OPT-7B models. This forgoes a plethora of open source models, experiments on which would have greatly strengthen the paper. Given that these LLMs are all from the Meta family of LLMs -- this raises some suspicion as to the generality of the proposed technique when different architectures or modeling assumptions are made.

**Questions:**

1. Do you have any results on other LLM families: Phi, MPT or Gemma?

**Limitations:**

Limitations are adequately addressed.

---

> ### Author Rebuttal · Authors · 2024-08-05
>
> We appreciate the effort and time of the reviewer (z7Cy). We are thrilled to receive positive feedback on the simplicity and scalability of our method, and the recognition of our efforts to characterise the gains. The reviewer's acknowledgement of our intuitive explanations for the results and the complementarity of our method with recent approaches like NEFTUNE is particularly gratifying. We would like to address the reviewer's valuable feedback as follows:
>
>
> > This forgoes a plethora of open source models, experiments on which would have greatly strengthen the paper. Given that these LLMs are all from the Meta family of LLMs -- this raises some suspicion as to the generality of the proposed technique when different architectures or modelling assumptions are made. (......) Do you have any results on other LLM families: Phi, MPT or Gemma?
>
> We appreciate the reviewer's concern about the generalizability of our results across different model families. In response to the reviewer's suggestion, we conducted additional experiments on additional models including Phi and Gemma, which show qualitatively similar results (see Table 2 in the rebuttal pdf). Our results show that our findings still hold with different language models.
>
> The meta family of LLMs are very standard transformers [1]. Indeed, the designers note that they tried to avoid innovating on the model architecture [2]. For example, the Llama model family has very standard features such as RoPE embeddings [3]. As such, we would not expect Llama family models to behave differently from other models.
>
> ### Reference:
>
> [1] Llama 2: Open Foundation and Fine-Tuned Chat Models.
>
> [2] The Llama 3 Herd of Models.
>
> [3] RoFormer: Enhanced Transformer with Rotary Position Embedding

---

### Author Rebuttal · Authors · 2024-08-05

We sincerely thank all reviewers for their thoughtful feedback and valuable suggestions. We are pleased that our work has been recognised for its novelty (`b1op`,`vkoa`), simplicity and scalability (`z7Cy`), extensive experiments (`b1op`, `kinw`, `vkoa`), and potential impact (b1op). Reviewers also appreciated our intuitive explanations for the results (`z7Cy`,`vkoa`), the complementarity of our method with recent approaches like Neftune (`z7Cy`), our well-written reasoning and thorough appendix (`b1op`), and the consistency of our results across diverse settings (`kinw`). In response to the reviewers' comments, we have conducted additional experiments and analyses, which we believe significantly strengthen our paper. Key updates include:

1. **Additional experiments**: We have tested our approach on additional model families, including Gemma (2B) and Phi-1.5 (1.3B) (`z7Cy`, `b1op`, `vkoa`). We also conducted experiments comparing high-quality curated datasets with randomly selected subsets, showing that our approach is robust across various data qualities (`b1op`, `vkoa`).

2. **Clarification on instruction definition**: We have provided detailed examples of what constitutes an "instruction" in our work, including static templates, dynamic user instructions, and formatting tokens (`kinw`).

3. **Loss analysis and overfitting**: We elaborated on our loss analysis experiments and BLEU score analysis to address concerns about potential overfitting analysis (`kinw`).

4. **Comparison with continued pre-training**: We clarified the differences between our approach and continued pre-training, highlighting the unique aspects of IM (`kinw`).

5. **Limitations**: We have clarified the location of our paper checklist (`vkoa`) and added notes on computational restrictions preventing experiments with very large models (60-70B parameters) (`b1op`, `vkoa`).

We believe these additions and clarifications address the main concerns raised by the reviewers and improve the overall quality and impact of our paper. We will incorporate these changes in our revised version.

---

### Decision · Program_Chairs · 2024-09-25

**Decision:**

Accept (poster)

**Comment:**

This paper proposes instruction modeling for training language models, which applies a function to the instruction and prompt part rather than solely to the output part. Experiments show that instruction modeling can improve model performance on both NLP tasks and open-ended generation benchmarks, with some key factors found that influence the effectiveness of the approach: the ratio between instruction and output lengths, and the quantity of training data.

Some reviewers raised concerns regarding to the limited model exploration and lacking the explaination of why applying loss to instructions works. The authors made clarifications in their rebuttal with some extra results, and all reviewers vote positive to this work. Overall, the proposed method is quite simple and scalable, and the results are impressive. This method would be of great interest to practitioners in the community.  Therefore, I recommend an "acceptance".